# Mortality and admissions for cerebrovascular and cardiovascular diseases after the Accumoli-Amatrice 2016 earthquake

Chiara Cadeddu[1], Mario Cesare Nurchis[2,3*], Carolina Castagna[3], Martina Sapienza[3], Rosaria Messina[3], Stefano Marchetti[4], Walter Ricciardi[3], Aldo Rosano[5]

1 Erasmus School of Health Policy and Management, Erasmus University Rotterdam, Rotterdam, Netherlands, 2 Department of Life Science, Health, and Health Professions, Università degli Studi Link, Rome, Italy, 3 Department of Health Science and Public Health, Section of Hygiene, Università Cattolica del Sacro Cuore, Rome, Italy, 4 Integrated System for Health, Social Assistance and Welfare, Italian National Institute of Statistics, Rome, Italy, 5 Unit of Statistics, National Institute for Public Policies Analysis (INAPP), Rome, Italy

* m.nurchis@unilink.it

## Abstract

### Objective

The aim of this study is to evaluate if the mortality and hospital admissions for cerebrovascular and cardiovascular diseases (CCVDs), with a specific focus on acute myocardial infarction (AMI), significantly increased within 12 months after the 2016 Accumoli-Amatrice earthquake in the 17 most hit municipalities.

### Methods

A diff-in-diff regression model was applied to the annual occurrences to assess the differences in the mortality and hospitalization for selected CCVDs before and after the Earthquake in the area most hit by the earthquake compared to a surrounding area. A focus analysis on acute myocardial infarction (AMI) was performed.

### Results

The analysis of mortality showed a significant increase in the death rate for AMI in the year immediately following the earthquake with +1.7 cases per 10,000 (95% CI: 1.53–1.87). A significantly increased number of hospital admissions and deaths for AMI was also noticed.

### Conclusions

Evaluating the epidemiology of CCVDs events related to natural disasters, might contribute to provide information for the population dealing with future events and can be used to improve preparedness in rescue and life-saving medical interventions during and after the earthquake.

**Data availability statement:** All relevant data are within the manuscript and its Supporting Information files.

**Funding:** The author(s) received no specific funding for this work.

**Competing interests:** The authors declare that they have no conflict of interest.

## Introduction

Cerebrovascular and cardiovascular diseases (CCVDs) are a class of diseases that involve the heart or blood vessels.

An estimated 17.9 million people died from CCVDs in 2019, representing 32% of all global deaths. Of these deaths, 85% are due to heart attack and stroke [1]. The most important behavioral risk factors for heart disease and stroke are unhealthy diet, physical inactivity, tobacco use and harmful use of alcohol. However, both acute and chronic stress can precipitate CCVDs [2] and the stress associated with natural disasters (i.e., extreme, sudden events caused by environmental factors) such as tsunami, hurricanes, and earthquakes may also have a negative impact on cardiac health [1–3].

Earthquakes in particular, provide a good example of naturally occurring acute stress, which represents the main trigger for cardiovascular events [4]. The stress caused by an earthquake is experienced differently by the individuals according to the type of damage and loss suffered [5]. The mechanism by which coronary artery damage occurs is due to the typical "fight or flight" adrenergic response triggered from the initial tremor and subsequent aftershocks [6].

Such responses are associated with the stimulation of the sympathetic nervous system and the release of catecholamines that increases oxygen demand (increased heart rate, blood pressure, and contractility of the ventricles) at the same time, having the potential to reduce oxygen delivery by constricting some vascular beds, causing coronary spasm, rupturing atherosclerotic plaques, and increasing the propensity to develop thrombi (increased platelet aggregation and reduced thrombolysis). This imbalance of the oxygen supply/demand equation can result in myocardial ischemia, myocardial infarction and contribute to lethal ventricular arrhythmias [7]. While the acute adrenergic response to the initial tremor and immediate aftershocks can explain the short-term cardiovascular effects of the earthquake, prolonged exposure to seismic activity over several months is also likely associated with the increase in the incidence of cardiovascular diseases. Chronic stress responses have been associated with sustained elevations in blood pressure, endothelial dysfunction, and an increased risk of thrombotic events [8,9]. Not every country in the world is equally susceptible to earthquakes. Italy represents one of the world's most earthquake-prone countries [10]. The Apennine mountains, in Central Italy, have the highest seismic hazard in Western Europe [11]. Such a territory has a long history of damaging earthquakes, including the most recent 2016 Accumoli-Amatrice earthquake, measuring 6 on the moment magnitude scale. The earthquake affected four regions of the Apennine land of Central Italy (i.e., Umbria, Marche, Abruzzo, and Lazio), six provinces (i.e., Perugia, Ascoli Piceno, Fermo, Rieti, L'Aquila, and Teramo) and in particular 17 municipalities. As of 26 August 2016, the official numbers of the Italian Civil Protection Department report that the earthquake caused the death of 297 people in the territory of 3 municipalities [12]. More than 365 injured had to be treated in hospitals, mainly in Rieti and Ascoli Piceno, while people with less serious injuries were treated on the spot. The initial earthquake was followed by an enormous number of aftershocks that proceeded for several months, felt across central Italy as

a whole. Even though Italy is a high-risk seismic country, only few Italian studies are currently available investigating the association between earthquakes and the occurrence of CCVDs [13–15]. Globally, there is growing evidence about an increased association between hospital admissions for CCVDs and earthquakes or major natural disasters.

According to Teng et al. 2017, rates of cardiovascular disease and myocardial infarction were increased in people living in severely damaged areas in the first year after the earthquake of Christchurch 2011, New Zealand [5].

The major findings of the study conducted by Nakamura A. et al., showed that the patient number of admissions for acute myocardial infarction was significantly increased and its severity was remarkably heightened after the Great East Japan Earthquake [16]; these results are strengthened by the work of Aoki et al. in which they showed how the weekly occurrences of CCVDs, including heart failure (HF), acute coronary syndrome (ACS), stroke, cardiopulmonary arrest (CPA), and pneumonia were all significantly increased after the Great East Japan Earthquake in 2011 compared with the previous 3 years [17].

In light of these considerations and given the paucity of Italian studies investigating this specific topic, the aim of this study is to evaluate if the mortality and hospital admissions for CCVDs, with a specific focus on Acute Myocardial Infarction (AMI), significantly increased within 12 months after the 2016 Accumoli-Amatrice earthquake in the 17 most hit municipalities.

## Methods

### Ethics statement

Ethical review and approval were waived for this study given the analyses of aggregated data. In this study, informed consent was not obtained as the data used was derived from aggregated administrative datasets provided by the Italian Ministry of Health. These datasets contained anonymized information, ensuring the privacy and confidentiality of individuals. The use of such aggregated data allowed for a comprehensive analysis at a population level and not at the individual one.

### Study design and case definition

A retrospective cohort study design was adopted to compare the hospitalization and mortality rates for CCVD occurring before the earthquake (i.e., August 26, 2016) to those that occurred after it. The exposure period considered went from August 24, 2016 (i.e., the day of Amatrice-Accumoli earthquake) to August 24, 2017, while the control period was defined from August 24, 2014 to August 23, 2016 for the hospitalizations and the period from 24 August 2011 to 23 August 2016 for the mortality. The length of the control periods was established in order to have a sufficient power of the study to estimate a risk ratio of hospitalization and mortality for CCVD of 1.4 associated with the exposure with a power of 80% and alpha error of 5%, considering the number of expected hospitalization and deaths for CCVDs based on occurrence in the same years observed in Italy and the size of the study population [18,19]. Furthermore, the length of the chosen time-span for the study is justified by the potential occurrence of stress-induced CCVD events due to the precarious living conditions after earthquake (i.e., makeshift shelter accommodations as tents and cars). Other factors should be considered in analyzing the occurrence of CCVD, like extreme weather events such as cold waves (as those occurred during the period analyzed). These examples serve to illustrate the types of stressors that may have prolonged the exposure of the affected population to adverse conditions. This rationale supports the chosen time span as a period during which the seasonal or cumulative effects of these environmental factors could have plausibly increased the risk of CCVD events.

We identified hospital admissions for acute CCVDs occurred from August 2014 to August 2017 and deaths with confirmed CCVD diagnosis occurred from August 2011 to August 2017, among individuals resident in the 17 most hit municipalities around Accumoli-Amatrice area (group A) and wherever admitted in hospitals in Italy, depicted in Fig 1. A control group (B) of municipalities was selected, including 45 municipalities surrounding the area most hit by the earthquake with similar characteristics of the group A municipalities, in terms of demographic characteristics, socioeconomic level and access to hospital care services [20]: Province of Ascoli Piceno: Comunanza, Cossignano, Force, Montalto delle Marche,

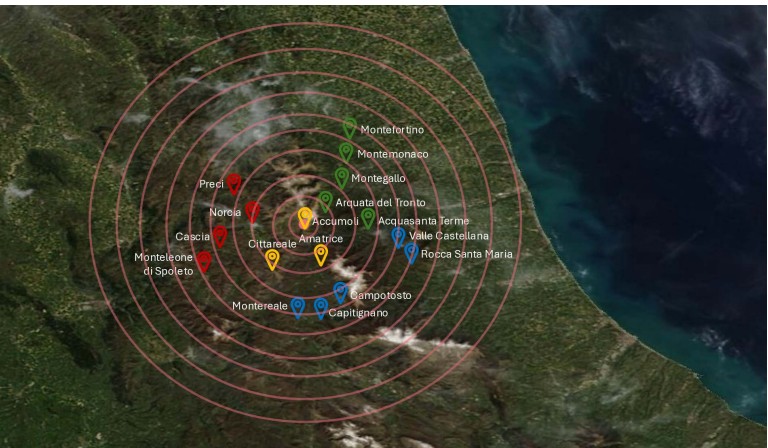

**Fig 1. Municipalities around Accumoli-Amatrice area. Green markers: Arquata del Tronto (AP), Acquasanta Terme (AP), Montegallo (AP), Montefortino (FM), Montemonaco (AP). Red markers: Preci (PG), Norcia (PG), Cascia (PG), Monteleone di Spoleto (PG). Yellow markers: Accumoli (RI), Amatrice (RI), Cittareale (RI). Blue markers: Montereale (AQ), Capitignano (AQ), Campotosto (AQ), Valle Castellana (TE) e Rocca Santa Maria (TE).**

Montedinove, Palmiano, Roccafluvione, Rotella, Venarotta; province of Fermo: Amandola; Province of Macerata: Acquacanina, Bolognola, Castelsantangelo sul Nera, Cessapalombo, Fiastra, Fiordimonte, Gualdo, Penna San Giovanni, Pieve Torina, Pievebovigliana, San Ginesio, Sant'Angelo in Pontano, Sarnano, Ussita, Visso; Province of Perugia: Cerreto di Spoleto, Poggiodomo, Sant'Anatolia di Narco, Scheggino, Sellano, Vallo di Nera; province of Rieti: Antrodoco, Borbona, Borgo Velino, Castel Sant'Angelo, Leonessa, Micigliano, Posta; province of Teramo: Cortino, Crognaleto, Montorio al Vomano; Province of Terni: Arrone, Ferentillo, Montefranco, Polino, Montalto delle Marche. Additional details on demographic, socioeconomic characteristics and access to hospital care facilities in the group A and group B municipalities are available in the supplementary materials and in S1 File, S1 File.

Fig 2 shows the number of earthquake aftershocks of magnitude greater than or equal to 4.5, recorded since Aug. 24, 2016, excluding those of magnitude less than 4.5 [21].

The 2016–2017 earthquake sequence recorded more than 118,000 earthquakes in Central Italy, of which 1,200 with magnitude between 3.0 and 3.9, 70 between 4.0 and 4.9, and 9 events with magnitude between 5.0 and 5.9. The main events with a magnitude of 6 or more occurred on August 24, 2016 in the Accumoli area and on October 30, 2016 in the Norcia area [22].

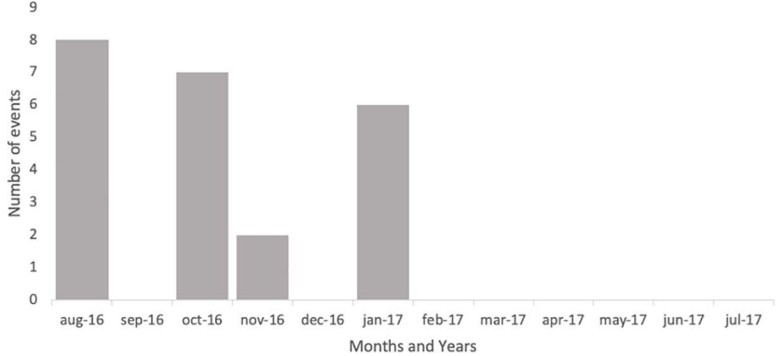

**Fig 2. Number of earthquakes with magnitudes higher than 4.5 after the Accumoli-Amatrice 2016 earthquake.**

Acute CCVDs were selected among those already investigated by evidence reported by other similar studies such as Teng et al. and Biffi et al. [5,15].

Then, CCVDs were defined according to the International Classification of Diseases, Ninth Revision, Clinical Modification (ICD-9-CM) for hospitalization data and the International Classification of Diseases, Tenth Revision (ICD-10) for mortality data. The included CCVDs and the related ICD codes are reported in S2 Table in S1 File.

Moreover, the settled status was defined according to the residency reported in the hospital discharge forms or in the cause of death certificates.

## Data sources

Information about the cardiovascular and cerebrovascular disease-related hospital admissions was retrieved by the Hospital Discharge Forms obtained by the Italian Ministry of Health.

Demographic data were provided by the Italian National Institute of Statistics (ISTAT) relative to the population living in the most-hit areas surrounding Accumoli-Amatrice (group A) and the control group of municipalities (group B) [23]. The number of deaths from all causes, by age group (i.e., <30; 30–39; 40–49; 50–59; 60–69; 70–79; 80–89; 90+), were obtained through elaborations on the vital statistics databases on causes of death by ISTAT. Mortality data were available on an annual basis, while hospitalization data were available on monthly basis. Data on temperatures were retrieved through the national system for the collection, processing and dissemination of climate data, created by the Italian Institute for Environmental Protection and Research (ISPRA) [24].

## Statistical analyses

Rates of cardiovascular and cerebrovascular disease-related hospital admissions, cardiovascular and cerebrovascular disease-related mortality and rate ratios were calculated.

First, we described the annual time trends in overall mortality rates as well as the monthly trends of hospitalization for CCVDs and AMI over the study period. A diff-in-diff design was adopted in order to compare the mortality rate and the hospitalization rate for selected CCVDs before and after the earthquake between the area most hit (group A of municipalities) and a control group of municipalities (group B). Four distinct regression models were used to estimate the effect of the earthquake on the mortality for CCVDs, mortality for AMI, hospitalization for CCVDs and hospitalization for AMI in the most hit area. The following variables were included into the models: the occurrence rate (mortality or hospitalization) for CCVDs (the outcome); (i) the group assignment variable (group A: most hit municipalities; group B: control group of municipalities); (ii) time point (before or after the earthquake), and (iii) the interaction term of these two factors, that is the difference-in-differences estimator. The equation of the difference-in-differences regression model was $Y = \alpha + \beta_1 \times (treatment) + \beta_2 \times (time\ point) + \beta_3 \times (treatment) \times (time\ point) + \beta_4 \times (sex) + \beta_5 \times (age\ class) + \varepsilon$, where Y is the outcome of interest, treatment is a dummy variable for group assignment (group A=1, control group B=0), and time point is a dummy variable for the time point (post-earthquake=1, pre-earthquake=0). Through the models the difference in the average change in mortality/hospitalization rate between the two groups of municipalities was estimated providing the effect of the earthquake on the occurrence of CCVDs and for AMI in the most hit group of municipalities.

Moreover, in order to avoid systemic biases, potential confounders as gender and age were included in the regression model. The time point variable assumed a value of 1 for the one-year period after the Earthquake (i.e., 24 August 2016 to 24 August 2017), and 0 for the period before the earthquake (from 24 August 2011 to 23 August 2016 for the analysis of mortality and from 24 August 2014 to 23 August 2016 for the analysis of hospitalization).

Furthermore, a focus analysis on Acute Myocardial Infarction (AMI) was also performed both observing simultaneously hospital admissions and the deaths due to AMI, including therefore sudden deaths that did not require a hospital admission. The observed number of hospitalized and dead patients for AMI after the earthquake in the most hit area was compared to the expected frequencies estimated on the basis of the occurrence from the previous 2 years and tested using $\chi 2$ –test.

The monthly mean temperatures of the exposure period were compared with those of the control period. Differences were tested through the t-test. Continuous variables are expressed as mean and standard deviation (SD). All statistical analyses were performed using Stata 17. Diff-in-diff analysis was conducted using the *didregress* function. The common trend assumption of the diff-in-diff model was tested using the *ptrend* function. All p-values were two-sided, and p-values of <0.05 were considered to be statistically significant.

## Results

The mortality rates in the study period showed a slight increase for AMI in the year immediately following the earthquake in the most hit area (Figs 3 and 4) (S3 and S4 Tables in S1 File).

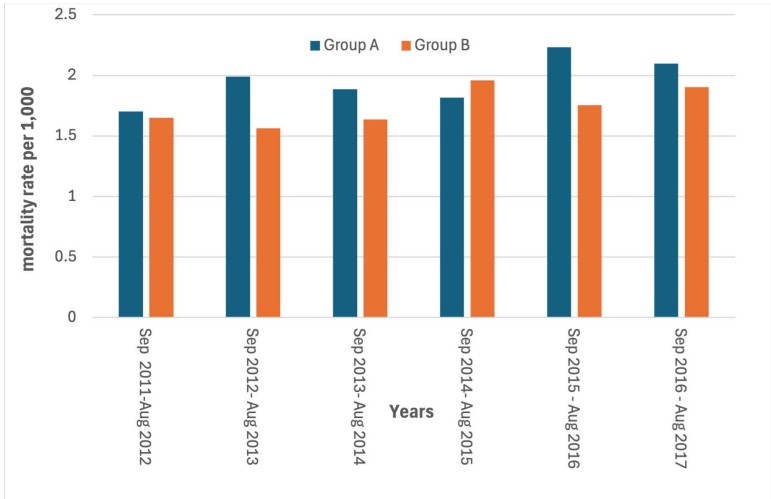

**Fig 3. Yearly mortality rates for CCCVDs by area. Period 2011-2017 (24 August – 23 August).**

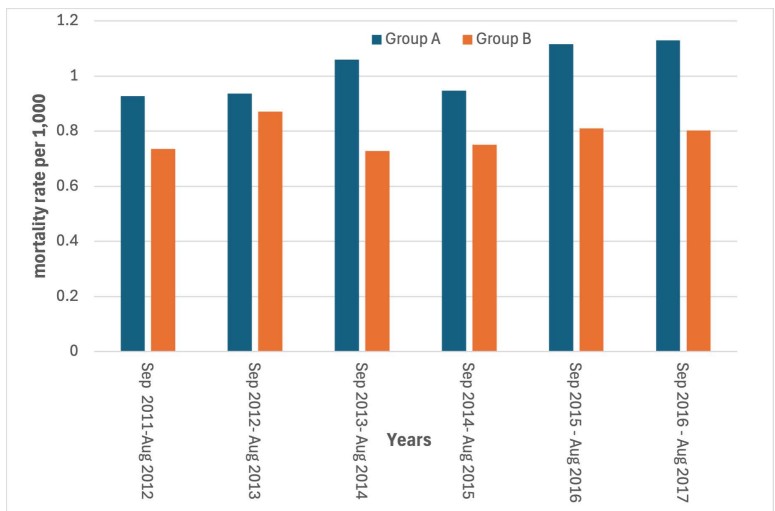

**Fig 4. Yearly mortality rates for AMI by area. Period 2011-2017 (24 August – 23 August).**

The series of hospitalization rates revealed an increase in the most hit area (group A) both for CCVDs (S5 Table in S1 File) and for AMI (S6 Table in S1 File) with a lag of 2–3 months after the earthquake (Figs 5 and 6).

Estimated rates through the diff-in-diff regression models are represented in Figs 7–10. The month when the earthquake occurred (August 2016) naturally divides the x-axis into two windows: post-earthquake (September 2016 – August 2017) and pre-earthquake period (September 2011-August 2016). An event study constructs DiD-type estimates in both windows, but they have different interpretations. The estimated rates in the post-treatment period reflect treatment (=earthquake) effect dynamics. The no-anticipation assumption of diff-in-diff design implies that average treatment effect (ATE) on Group A before the earthquake is equal to zero, which implies the parallel trend in the rates distribution of Group A and Group B. The findings of the regression model on the risk of dying for CCVDs in the year following the earthquake in the most hit area, adjusted for age and gender, showed no significant increase in the exposure period with respect to control group of municipalities with an estimated effect on the population in the most hit area of +2.14 cases per 10,000

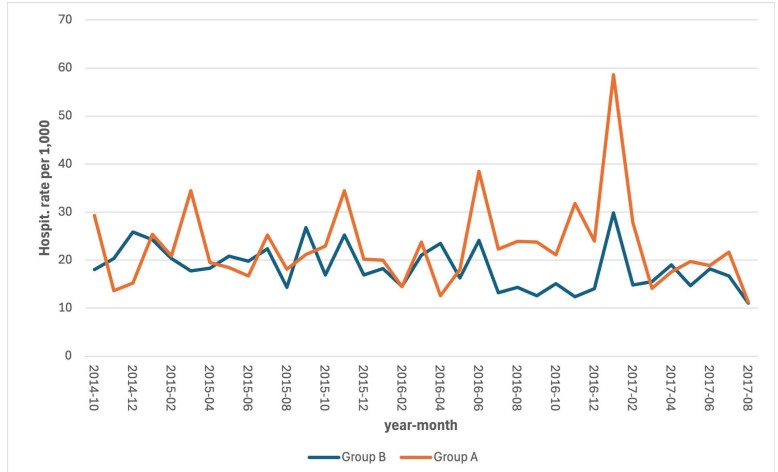

**Fig 5. Monthly hospitalization rates for CCVDs by area per 1,000 persons. Period: September 2014 – August 2017.**

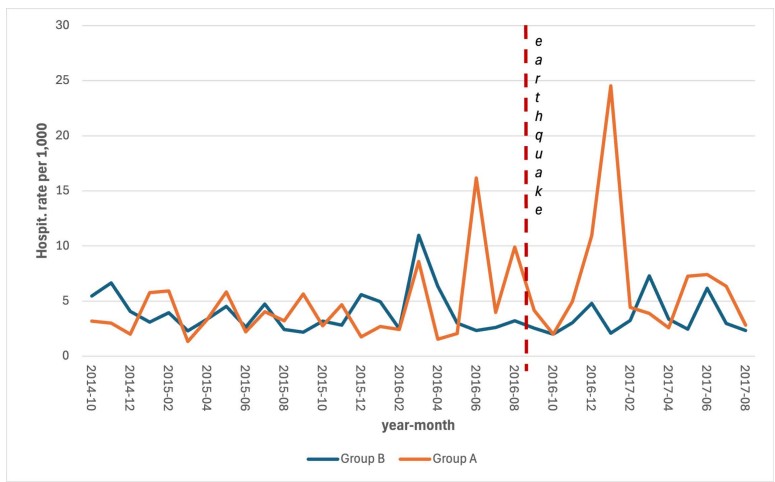

**Fig 6. Monthly hospitalization rates for AMI by area, per 1,000 persons. Period: September 2014 – August 2017.**

(95% Confidence Interval: −0.47–4.75). Conversely, a significant increase in the risk of dying from AMI in the year following the earthquake was observed with an estimated average effect on the population in the most hit area of +1.7 cases per 10,000 (95% Confidence Interval: 1.53–1.87). (Figs 7 and 8).

The parallel-trends assumption was tested and no deviation from the assumption was found both for CCVDSs (p = 0.15) and AMI (p = 0.09).

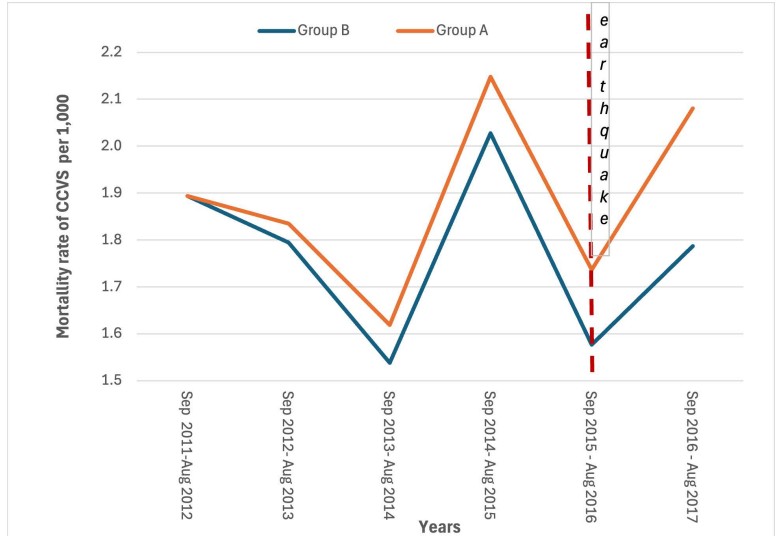

**Fig 7. Linear trends estimated through the diff-in-diff regression models in the most hit area (group A) and in the control group of municipalities (group B) . Mortality rates of CCVDs. Period 2011-2012 to 2016-2017.**

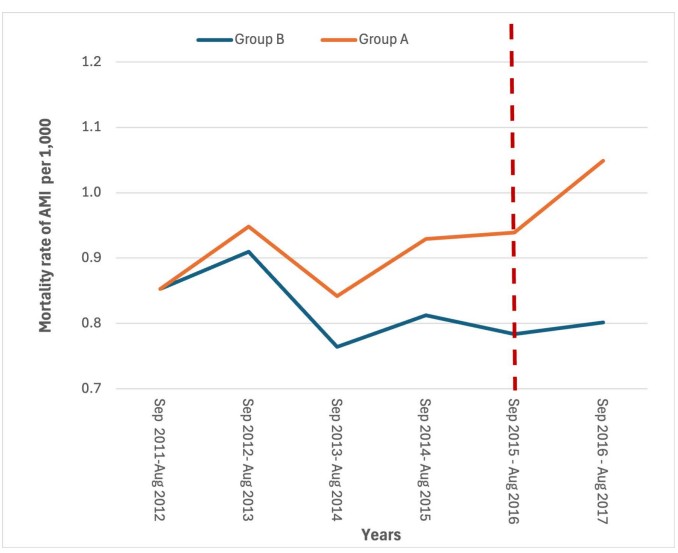

**Fig 8. Linear trends estimated through the diff-in-diff regression models in the most hit area (group A) and in the control group of municipalities (group B). Mortality rates of AMI. Period 2011-2012 to 2016-2017.**

The findings of the regression model on the hospitalization for CCVDs in the months following the earthquake in the most hit area, adjusted for age and gender, showed an increase, though not statistically significant, in the exposure period with respect to control group of municipalities with an estimated average effect on the population in the most hit area of +6.72 cases per 10,000 (95% Confidence Interval:-19.04–32.48), as well as for IMA, with +2.91 cases per 10,000 (95% Confidence Interval: −5.94–11.77) (Figs 9 and 10).

The parallel-trends assumption was tested and no deviation from the assumption was found both for CCVDSs (p = 0.75) and AMI (p = 0.07).

Details on observed mortality and hospitalization rates for CCVD and AMI, as well as linear trends estimated rates are available in the supplementary material, S2–S5 Tables in S1 File.

Furthermore, as pointed out in Table 1, a significantly increased number of hospital admissions and "admissions and deaths" for AMI was noticed in the most hit area in the year following the earthquake. Such an increase can be observed

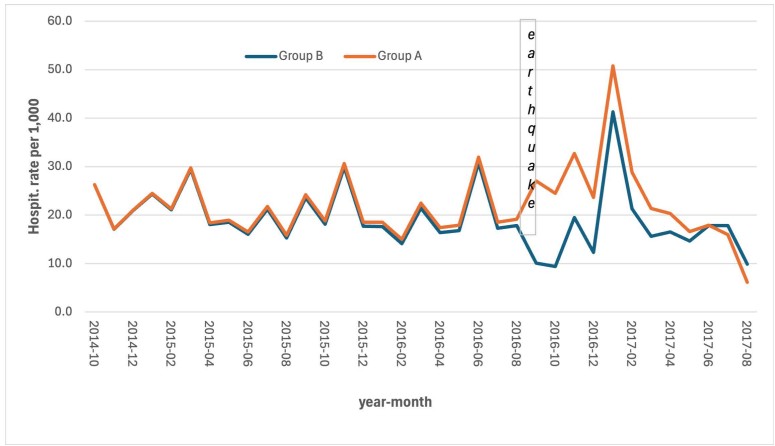

**Fig 9. Linear trends estimated through the diff-in-diff regression models in the most hit area (group A) and in the control group of municipalities (group B). Hospitalization rates of CCVDs. Period 2014-September – 2017- August.**

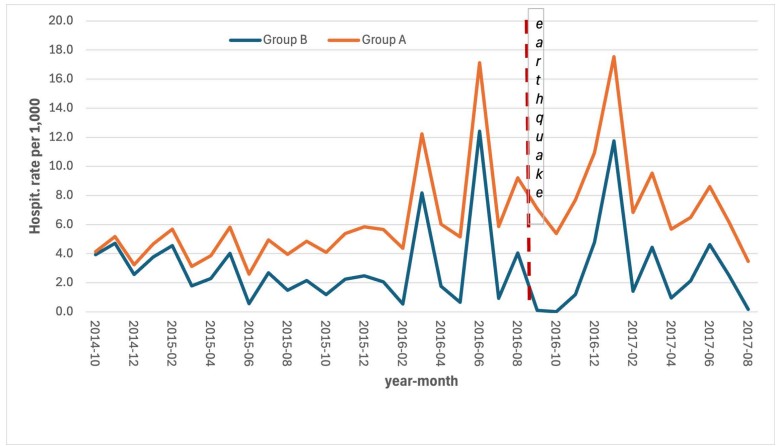

**Fig 10. Linear trends estimated through the diff-in-diff regression models in the most hit area (group A) and in the control group of municipalities (group B). Hospitalization rates of AMI. Period 2014-September – 2017- August.**

**Table 1. Hospital admissions and deaths for AMI two years before and one year after the earthquake by age and sex. Comparisons between observed events (one year after the EQ) and expected one, estimated through the admissions occurred 2 years before the EQ.**

| Hospital Admissions + Deaths 2 years before the earthquake | | | Hospital Admissions + Deaths 1 year after the earthquake | | | Total observed cases | Total expected cases | Observed / Expected | Chi square values | p |
|---|---|---|---|---|---|---|---|---|---|---|
| Age class | M | F | T | M | F | T | | | | | |
| 30-39 | 1 | 0 | 1 | 1 | 0 | 1 | 1 | 0.5 | 2.0 | 0.50 | 0.48 |
| 40-49 | 4 | 1 | 5 | 3 | 1 | 4 | 4 | 2.5 | 1.6 | 0.90 | 0.34 |
| 50-59 | 17 | 5 | 22 | 8 | 3 | 11 | 11 | 11 | 1.0 | 0.00 | 1.00 |
| 60-69 | 25 | 4 | 29 | 16 | 4 | 20 | 20 | 14.5 | 1.4 | 2.09 | 0.15 |
| 70-79 | 23 | 9 | 32 | 20 | 11 | 31 | 31 | 16 | 1.9 | 14.06 | 0.00 |
| 80-89 | 13 | 26 | 39 | 9 | 10 | 19 | 19 | 19.5 | 1.0 | 0.01 | 0.91 |
| 90+ | 6 | 14 | 20 | 4 | 6 | 10 | 10 | 10 | 1.0 | 0.00 | 1.00 |
| Total | 89 | 59 | 148 | 61 | 35 | 96 | 96 | 74 | 1.3 | 6.54 | 0.01 |

*Abbreviations: M, males; F, females; T, total; p, p-value.*

in all the age groups of the population living in the most-hit areas, but especially in the 60–69 and 70–79 age groups. Overall, in the year after the earthquake a number of occurrences (i.e., hospital admission and deaths) of AMI higher than what would have been expected (Observed/Expected ratio = 1.3, *p* = 0.01) was observed.

No long outbreaks of cold temperatures were observed in the four months after the earthquake and the monthly mean temperatures of the exposure period were not significantly different from those of the control period. The only exceptions were February, March, June and August, when significantly higher temperatures were observed (+2.5° in February; + 1.8° in March; + 2.6° in June and +3.4 in August), and January with significantly lower temperature (- 3.2°).

## Discussion

As far as we know, this study represents the first one conducted on the association between major earthquakes and hospitalizations and deaths for CCVDs in Italy. It found a statistically significant increase in the mortality rate for AMI and, although not statistically significant, for hospitalizations due to CCVDs and related mortality. Particularly, the series of hospitalizations rates showed a spike in the most hit area, both for CCVDs and for AMI, with a lag of 2–3 months which matches with the significantly lower temperature recorded in January. These increases have occurred at a time when the rest of the country has seen constant decreases both in mortality rates for CVD and hospitalization for AMI. The mortality rates for coronary heart disease decreased from 11.5 per 10,000 in 2011 to 9.0 per 10,000 in 2017 [25]; the hospitalization for AMI decreased from 132,896 accesses in 2015–129,763 in 2017 [26].

The observed increase in hospitalization rates in Group A prior to the earthquake event can be largely attributed to the natural aging of the population, a key determinant of healthcare utilization over time. Furthermore, the peak recorded in June 2016 coincides with an extreme heat wave, which likely contributed to excess hospitalizations in both groups, particularly among the elderly.

Another expected finding was the significant increase after the earthquake of hospital admissions and deaths for AMI (simultaneously considered) which were observed particularly in the 60–69 and 70–79 age groups given the tendency of this condition to affect mainly older people [27–29]. In particular, in the year following the earthquake we found 7 deaths out of 83 admissions and 13 deaths before the arrival in hospital, that is 13.5% of deaths before the arrival in hospital and a case fatality ratio in hospital of 8.4%. We could not have sex-stratified data from our sources, but it can be generally said that these results are in line with those by Asaria et al. [30], which in a study on hospitalization and mortality data for myocardial infarction in England found, after age-standardization, 15.0% of events in women and 16.9% in men resulted in death before hospitalization, and hospital case fatality of 10.8% in women and 10.6% in men.

The association between the risk of CCVDs and earthquakes has already been investigated in Italy [13,15,31] and in other countries around the world such as New Zealand (5), Japan, USA, and China [16,17,32–39]; though, the data available in Italy are currently pretty scarce.

A case report discussed the occurrence of ventricular fibrillation triggered by the Accumoli-Amatrice earthquake and a multicentric retrospective cohort study analyzed the influence of earthquakes on the occurrence of aortic aneurysm ruptures [15,30].

Similarly to the findings highlighted by our study, both these last two cited studies found an increased occurrence of cardio- and cerebro-vascular events after the earthquake.

Our findings align also with several studies from specialized disaster medicine and cardiology literature. In disaster medicine, Huang et al. documented similar patterns of increased cardiovascular events following the 2008 Wenchuan earthquake in China, emphasizing the importance of sustained medical surveillance in affected areas [40]. From a cardiological perspective, Takegami et al. demonstrated that earthquake-induced stress leads to specific pathophysiological changes including increased sympathetic activity and endothelial dysfunction, which may persist for months after the initial event [41]. More recently, Yamaoka-Tojo and Tojo emphasized the diverse cardiovascular sequelae of natural disasters, ranging from acute coronary syndromes to arrhythmias and stress-induced cardiomyopathy, and highlighted the need for early risk stratification and multidisciplinary interventions to mitigate these risks [42]. Complementing this, disaster-specific guidelines jointly issued by the Japanese Circulation Society and other professional bodies offer targeted recommendations for the management of cardiovascular conditions during and after natural disasters, underscoring the importance of maintaining continuity of care and emergency preparedness [43].

Considering the main possible implications, it is essential to encourage a critical reflection at the four levels (i.e., macro, meso, micro and nano) characterizing the decision-making process. At the macro level, it is crucial to set up a national plan of action for the preparedness and management of natural disasters. At the meso level, it could be desirable to create a territorial network among general practitioners and other medical specialists in the identification, monitoring and early management of individuals at risk in case of natural disasters. At the micro level, it is necessary to train healthcare professionals with dedicated courses. Moreover, hospitals in areas most at risk should consider the development of guidelines and targeted clinical pathways for handling increases in myocardial infarctions and other cardiac events that are associated with earthquakes. At nano level, all the individuals and especially those living in the most earthquake prone areas should be informed about the typical signs and symptoms of AMI over time and what to do as soon as they appear.

Our study showed that modern public health cannot be separated from the values and the milestones of planetary health, demonstrating that investing in a brand-new public health ecological approach [44] might ease the implementation of strategies and roadmaps to foster resilience and adaptation of ecosystems (including health/social/political/economic systems) [45]. By way of example, in Japan or in other world's most earthquake-prone countries over the years a real process of adaptation of urban architecture to the territory has been carried out, resulting in the broad concept of seismic engineering [46]. Moreover, education and training initiatives on natural disasters were provided for years among the population living in seismic areas [47]. Through this system of "preparedness", it is possible to mitigate the impact of natural disasters on human health.

On the contrary, Italy still lacks a culture of prevention and disaster preparedness and relies mostly on hurried tactics in crisis response rather than strategic planning to deal with emergencies. It is important to stimulate thoughts in this sense in order to minimize the damages on human health from natural disasters. In planning and implementing preparedness policies, central and regional governments should also consider the immediate and long-term stress-induced cardiovascular consequences due to logistic issues of new precarious housing [48].

Hence, through a deeper understanding of the impact of natural disasters on human health, as in the case of such conditions, we may have the opportunity to learn how to properly manage natural disasters and better identify the needs of affected populations. In fact, thanks to studies such as this one, it may be possible to comprehend how and when to intervene and prevent the occurrence of AMIs and other acute diseases in which psycho-physical stress plays a key role.

Our study should be considered in the light of some limitations. This is a retrospective cohort study describing the association between exposure to intervention and outcome. For privacy reasons, it was not possible to collect individual data as well as some information, such as the exact date of discharge or death, but only the year of death and month of the hospital admission. In addition, due to the impossibility of obtaining some more accurate information, confounders related to a direct measure of socio-economic status and a history of past illnesses which increase the risk of CCVDs may have caused some systematic biases. A key limitation of this study is the reliance on aggregated data, which may reduce the granularity of the analysis and increase the risk of ecological bias. This constraint was due to privacy regulations that limit access to individual-level health data for academic researchers in our country. As a result, while our findings are robust at the population level, caution is warranted in extrapolating them to individual-level associations. Another caveat could be represented by the fallacy given by choosing the individuals' residence, that is the formal address indicated by each person for governmental purposes, which in some cases does not overlap with the effective place where a person actually lives and spends most time for job, personal, familiar or other reasons. Consequently, some of the individuals may have not been really exposed to the event. A potential limitation related to some recognized additional risk factors for AMI could be identified in the unusual cold registered in January after the earthquake which could have contributed to a part of hospitalizations and/or deaths and the inconvenient home conditions of many people after the event, i.e., camping tents, caravans and containers, which could have caused further stress and impact on the cardiovascular well-being. However, we were able to control this potential limitation due to stress cold-related by using a quasi-experimental design, the diff-in-diff, that allowed us to consider a control group of municipalities that were likely to be affected by the same environmental exposure. Another limitation concerns the absence of a negative control analysis, which may weaken causal inference. However, the steady decreases in both mortality rates for CCVD (in particular ischemic heart diseases) and in hospitalization for AMI in the rest of the country over the study period constitute good evidence to support the findings.

Oppositely, the strengths of the study consist in the originality of its research question and purpose, and in the longitudinal comparison of the population. In fact, thanks to this methodological approach, it was possible to observe the same population over a period of time. A further strength is the diff-in-diff design adopted, which allowed estimation of the effect of the event controlling for possible trend in the outcome or other confounding factors, such as cold temperatures, which equally affected the most hit area and the control group of municipalities. The validity of the design is confirmed by the results of the tests for parallel trends, that was largely not significant for CVD, both for mortality and for hospitalization, while for AMI, even though the graphical test does not indicate a clear parallel trend, the statistical tests support the hypothesis parallel trend assumption is not violated. Unfortunately, the same quasi-experimental design could not be applied to the analysis of combined hospital and mortality data for IMA because of the few observation points. On the other hand, the evidence from distinct mortality and hospitalization data on CVD and AMI, where a control group was considered, provides reliable confidence on the same patterns found in analysis of combined data of hospitalizations and mortality for AMI. Moreover, differently from other similar evidence available in the scientific literature, an overall analysis of AMI, taking into account both hospital admissions and mortality data, was conducted. In this regard, the inclusion of data about Hospital Discharge Forms but also governmental statistics about causes of death gave a more complete overview of the phenomena, because both deaths occurred in hospitals as well as those occurred before reaching hospitals were considered. Finally, it should be noticed that the ISTAT guidelines for the statistics about causes of death [49] indicate that if the duration of myocardial infarction is not declared, it is considered chronic only if specified or if it is described of longer duration but without years of illness. Consequently, all the other infarctions are counted as AMI. Furthermore, the pre-earthquake heterogeneity between our study and control groups warrants careful consideration when interpreting our results. Baseline differences in population age structure, socioeconomic status, healthcare accessibility, and pre-existing cardiovascular disease prevalence may have influenced both vulnerability to earthquake-related stress and healthcare utilization patterns. These pre-existing disparities underscore

the complex interplay between social determinants of health and disaster outcomes. as noted by Cornelius et al. (2022) [50] and Davis et al. (2010) [51]. It is important to note that our study, while consistent with international literature, presents less detailed pathophysiological mechanisms and less granular data compared to Japanese studies. Japanese research has benefited from their extensive experience with seismic events, well-established disaster medicine infrastructure, and sophisticated health monitoring systems specifically designed for earthquake scenarios. The pathophysiological mechanisms underlying earthquake-induced cardiovascular events involve complex interactions between acute sympathetic activation, inflammatory processes, endothelial dysfunction, and hemodynamic changes that may persist for months after the initial event. These include sustained elevations in catecholamines, increased oxidative stress, alterations in platelet function, and vascular inflammation that collectively increase cardiovascular risk. Future Italian studies would benefit from more detailed clinical data collection, including biomarkers of stress and inflammation, ambulatory blood pressure monitoring, and cardiac imaging data to better characterize these pathophysiological processes.

Given the paucity of studies conducted in the Italian scenario on this topic, further research is advisable to fill the data gap we could not collect due to privacy reasons. Further studies could conduct difference-in-difference experiments to compare, pre and post the event, the investigated hit population with respect to the overall Italian one.

## Conclusions

The last two decades are seeing a frightening increase in natural disasters around the world, which are supposed to be linked to a possible dramatic increase of seismic activity [52,53]. Evaluating with additional studies the long-term CCVDs events related to natural disasters, including earthquakes, along with our results, will contribute to provide useful information for the population dealing with major disasters in the future and can be used to improve preparedness in rescue and life-saving medical interventions during and after the earthquake.

## Supporting information

**S1 File. This file contains Supplementary Tables S1–S6 and Supplementary Figures S1–S2.**
(DOCX)

## Acknowledgments

We sincerely thank Prof. Americo Cicchetti, Mr. Gianrico Di Fonzo and Mr. Claudio Colandrea from the Italian Ministry of Health for their support in sharing hospitalizations data.

## Author contributions

**Conceptualization:** Chiara Cadeddu, Walter Ricciardi, Aldo Rosano.

**Data curation:** Mario Cesare Nurchis, Carolina Castagna, Martina Sapienza, Rosaria Messina, Aldo Rosano.

**Formal analysis:** Aldo Rosano.

**Investigation:** Chiara Cadeddu, Carolina Castagna, Martina Sapienza.

**Methodology:** Chiara Cadeddu, Mario Cesare Nurchis, Aldo Rosano.

**Resources:** Mario Cesare Nurchis, Carolina Castagna, Aldo Rosano.

**Software:** Aldo Rosano.

**Supervision:** Chiara Cadeddu, Stefano Marchetti, Walter Ricciardi, Aldo Rosano.

**Validation:** Chiara Cadeddu, Stefano Marchetti, Walter Ricciardi.

**Visualization:** Chiara Cadeddu, Mario Cesare Nurchis, Stefano Marchetti, Walter Ricciardi.

**Writing – original draft:** Chiara Cadeddu, Mario Cesare Nurchis, Carolina Castagna, Martina Sapienza, Rosaria Messina, Aldo Rosano.

**Writing – review & editing:** Chiara Cadeddu, Mario Cesare Nurchis, Carolina Castagna, Martina Sapienza, Rosaria Messina, Stefano Marchetti, Walter Ricciardi, Aldo Rosano.

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
