## [Decision Letter · Decision Letter 0]

PONE-D-23-15690Mortality and admissions for cardiovascular diseases after the Accumoli-Amatrice 2016 earthquakePLOS ONE

Dear Dr. Nurchis,

Thank you for submitting your manuscript to PLOS ONE. After careful consideration, we feel that it has merit but does not fully meet PLOS ONE’s publication criteria as it currently stands. Therefore, we invite you to submit a revised version of the manuscript that addresses the points raised during the review process.

We look forward to receiving your revised manuscript.

Kind regards,

Francesco Branda, Ph.D.

Academic Editor

PLOS ONE

Journal Requirements:

2.  In the online submission form, you indicated that "The datasets generated during and/or analyzed during the current study are available from the corresponding author on reasonable request." 

3. We note that Figure 1 in your submission contain map/satellite images which may be copyrighted. All PLOS content is published under the Creative Commons Attribution License (CC BY 4.0), which means that the manuscript, images, and Supporting Information files will be freely available online, and any third party is permitted to access, download, copy, distribute, and use these materials in any way, even commercially, with proper attribution. For these reasons, we cannot publish previously copyrighted maps or satellite images created using proprietary data, such as Google software (Google Maps, Street View, and Earth). For more information, see our copyright guidelines: http://journals.plos.org/plosone/s/licenses-and-copyright.

Additional Editor Comments:

The article needs major corrections. In particular, the most critical point is the statistical approach taken by the authors. In addition, a revision of the English language is required to improve the fluency of the text and a more detailed explanation of the selection of ICD codes in hospital data.

Reviewers' comments:

Reviewer's Responses to Questions

**Comments to the Author**

1. Is the manuscript technically sound, and do the data support the conclusions?

Reviewer #1: Yes

Reviewer #2: Partly

2. Has the statistical analysis been performed appropriately and rigorously? 

Reviewer #1: Yes

Reviewer #2: No

3. Have the authors made all data underlying the findings in their manuscript fully available?

Reviewer #1: Yes

Reviewer #2: No

4. Is the manuscript presented in an intelligible fashion and written in standard English?

Reviewer #1: Yes

Reviewer #2: Yes

5. Review Comments to the Author

Reviewer #1: The paper addresses a very interesting topic that has been little addressed in the scientific literature. The work is excellent and the conclusions interesting.

Below are my minor comments:

1) A proof reading of English can help make the paper more fluent. For example, the first sentence "Cardiovascular diseases (CVD) is a class of diseases that involve the heart or blood vessels." can be better rendered without repeating disease.

2) I suggest to deeply explain how the authors selected ICD among the hospital data. Although they are listed in Table S1, more details explaining what and way is included could be useful.

3) Quality of Figure 2 to Figure 4 could be improved.

After this minor correction I think that the paper could be suitable for publication in Plotone.

Reviewer #2: I would like to thank you for providing me the opportunity to read and provide a review of this paper.

I've found interest in reading the paper even if the research question explored is not particularly new given the existing literature on earthquakes and CVDs worldwide (e.g. Japan and New Zealand). I have a major concern regarding the statistical approach adopted by the authors.

If I got it right, the methodology adopted by the authors is basically a before/after comparison of those individuals with residence in the 17 selected municipalities that were partially or totally affected by the Accumoli-Amatrice earthquake. The control period considered is basicallly the pre-earthquake period: from August 24, 2014 to August 23, 2016.

However this approach, as clearly discussed by the authors, is subject to some relevant limitations. In particular, results may be biased by “additional risk factors for AMI” such as “[…] unusual cold registered on January after the earthquake which could have contributed to a part of hospitalizations and/or deaths and the inconvenient home conditions of many people after the event, i.e. camping tends, caravans and containers, which could have caused further stress and impact on the cardiovascular wellbeing”.

However, these kind of biases may be eliminated if the authors would have adopted a different econometric technique among those belonging to the Counterfactual Impact Evaluation (CIE) family that better suits the research question investigated (for example a Difference in Difference model?). Including also data on individuals that were resident in municipalities closed to the earthquake but that were not affected by it because they were not in proximity of the epicentre as control group -for example- may help in taking into account any simultaneous risk factor that may have affected the population in the considered periods before and after the sudden earthquake event.

I wonder why the authors limited their analysis to the simple before/after approach while other CIE tecniques would better fit in this context. I haven't found any discussion about it in the actual version of the paper while I personally think that the choice of the appropriate identification strategy is crucial in this context.

6. PLOS authors have the option to publish the peer review history of their article (what does this mean? ). If published, this will include your full peer review and any attached files.

**Do you want your identity to be public for this peer review?** For information about this choice, including consent withdrawal, please see our Privacy Policy .

Reviewer #1: No

Reviewer #2: No

---

## [Author Response · Author response to Decision Letter 1]

25 Apr 2024

Reviewer 1

The paper addresses a very interesting topic that has been little addressed in the scientific literature. The work is excellent and the conclusions interesting.

We really thank the reviewer for the valuable suggestions. We addressed all your comments improving the scientific soundness of the manuscript.

1) A proof reading of English can help make the paper more fluent. For example, the first sentence "Cardiovascular diseases (CVD) is a class of diseases that involve the heart or blood vessels." can be better rendered without repeating disease.

Thank you for the comment. We improved the readability of the manuscript. Please, see modified paper.

2) I suggest to deeply explain how the authors selected ICD among the hospital data. Although they are listed in Table S1, more details explaining what and way is included could be useful.

Thank you for the valuable comment. We added an explanation about how we selected ICDs. Please, see modified paper.

3) Quality of Figure 2 to Figure 4 could be improved.

We thank the reviewer for the comment. As for a new suggested analysis, we deleted the previous figures. However, we added new ones whose quality is also higher.

Reviewer 2

I would like to thank you for providing me the opportunity to read and provide a review of this paper.

I've found interest in reading the paper even if the research question explored is not particularly new given the existing literature on earthquakes and CVDs worldwide (e.g. Japan and New Zealand).

We really thank the reviewer for the valuable suggestions. We are very grateful for their appreciation and their precious comments, that we strictly followed.

I have a major concern regarding the statistical approach adopted by the authors.

If I got it right, the methodology adopted by the authors is basically a before/after comparison of those individuals with residence in the 17 selected municipalities that were partially or totally affected by the Accumoli-Amatrice earthquake. The control period considered is basicallly the pre-earthquake period: from August 24, 2014 to August 23, 2016.

However this approach, as clearly discussed by the authors, is subject to some relevant limitations. In particular, results may be biased by “additional risk factors for AMI” such as “[…] unusual cold registered on January after the earthquake which could have contributed to a part of hospitalizations and/or deaths and the inconvenient home conditions of many people after the event, i.e. camping tends, caravans and containers, which could have caused further stress and impact on the cardiovascular wellbeing”.

However, these kind of biases may be eliminated if the authors would have adopted a different econometric technique among those belonging to the Counterfactual Impact Evaluation (CIE) family that better suits the research question investigated (for example a Difference in Difference model?). Including also data on individuals that were resident in municipalities closed to the earthquake but that were not affected by it because they were not in proximity of the epicentre as control group -for example- may help in taking into account any simultaneous risk factor that may have affected the population in the considered periods before and after the sudden earthquake event.

I wonder why the authors limited their analysis to the simple before/after approach while other CIE tecniques would better fit in this context. I haven't found any discussion about it in the actual version of the paper while I personally think that the choice of the appropriate identification strategy is crucial in this context.

We thank the reviewer for the valuable comment. Following your kind suggestion, we decided to adopt the suggested counterfactual impact evaluation approach. For this aim we collected relevant data also from a group of municipalities in the area surrounding the epicentre of the earthquake but with limited effects compared to the most hit area, previously selected. Hence the analysis was conducted through Difference-in-Differences (DID) regression models, mitigating the effects of extraneous factors (e.g. cold temperatures and other environmental factors) and selection bias. The results of the DID regression confirmed the increase in the post-earthquake period of the occurrence of Acute myocardial infarction in the most hit area compared with the control group of municipalities.

---

## [Decision Letter · Decision Letter 1]

PONE-D-23-15690R1Mortality and admissions for cerebrovascular and cardiovascular diseases after the Accumoli-Amatrice 2016 earthquakePLOS ONE

Dear Dr. Nurchis,

Thank you for submitting your manuscript to PLOS ONE. After careful consideration, we feel that it has merit but does not fully meet PLOS ONE’s publication criteria as it currently stands. Therefore, we invite you to submit a revised version of the manuscript that addresses the points raised during the review process.

The review of the manuscript raises several questions, particularly regarding the use of the DID model and the parallel trend hypothesis. Although the authors conducted statistical tests to evaluate it, the data show an increase in the hospitalization rate in Group A even before the earthquake event, as evidenced in Figures 4a, 4b, and 6b, which may compromise the validity of the analysis. More detailed explanations are needed on this point and on the biological plausibility of the observed delayed effects, considering the possible role of aftershocks. A negative control with random interventions and the inclusion of context-specific time variables is suggested to improve causal interpretation. In addition, the tables do not clearly show the main results of the DID, which should be included in the text or supplementary materials. Some methodological issues, such as the selection of analysis periods and the inclusion of age and gender in the models, require clarification. Finally, it is suggested that the figures be improved, precise references be added, and the interpretation of the results be expanded, with special attention to data structure and visual presentation.

We look forward to receiving your revised manuscript.

Kind regards,

Francesco Branda, Ph.D.

Academic Editor

PLOS ONE

Reviewers' comments:

Reviewer's Responses to Questions

**Comments to the Author**

1. If the authors have adequately addressed your comments raised in a previous round of review and you feel that this manuscript is now acceptable for publication, you may indicate that here to bypass the “Comments to the Author” section, enter your conflict of interest statement in the “Confidential to Editor” section, and submit your "Accept" recommendation.

Reviewer #3: (No Response)

2. Is the manuscript technically sound, and do the data support the conclusions?

Reviewer #3: Partly

3. Has the statistical analysis been performed appropriately and rigorously? 

Reviewer #3: Yes

4. Have the authors made all data underlying the findings in their manuscript fully available?

Reviewer #3: Yes

5. Is the manuscript presented in an intelligible fashion and written in standard English?

Reviewer #3: No

6. Review Comments to the Author

Reviewer #3: Thank you for sharing a very interesting manuscript. This study evaluated the association between earthquake exposure and cardiovascular mortality and hospitalization using difference-in-difference (DID) framework. I was only able to see the revised version of the manuscript and I believe there are several major and minor issues regarding this manuscript.

Major comment

: I agree with the previous reviewer 2’s comment, and changing the overall analyses using the DID framework seem reasonable. Major assumption of the DID analysis is parallel trend (as authors well indicated), and authors have evaluated it using statistical testing. However, basic evaluation of the assumption can be done with simple plotting of Group A and B (such as in Figure 4).

In the Figures 4a 4b 6b, we can clearly see the increase in hospitalization rate in Group A even before the earthquake event (2016 06, 07, 08, 09). How could authors explain these issues? Even a statistical testing for the common trend assumption shows marginally significant results, indicating that the potential difference of the trend before the earthquake event between Groups A and B. If authors shorten their analysis period for the common trend testing, I am bit worried that they might have significant p-values. I believe reasonable and clear explanations are needed for this issue before conducting DID analysis.

: More explanation for the biological plausibility of authors’ findings are needed. Current explanation in the introduction section focused mainly on the acute association (line 61-67, increase of stress right after the earthquake). Authors should explain why there are some lagged effects, and the effects of earthquake lasted for several months. Is it because of the long-lasting aftershocks? (BTW, I cannot see the figure 2 and hard for me to check the frequency of the aftershocks.)

: To increase the possibilities of causal interpretation, I suggest authors to add some negative control analysis by using different intervention timing (some random point before the earthquake) or disease outcomes irrelevant to earthquake exposure.

: Precise referencing is needed. For example: line 59-60, 323-334, and many others.

: Current table 1 is not showing the results of the DID analysis. Please include tables showing the pre-earthquake rate, post-earthquake rate and DID estimates (main findings of this study). There should be tables corresponding to Figure 3-6 (please consider using the supplementary material).

: Please add more explanation for selecting different time frame for mortality and hospitalization. Authors presented it was based on the power calculation but not sure how this can be applied to the DID framework (Not familiar with DID studies using it). In addition, please change reference 15 16 to journal paper if it is possible. Line 121-124 needs more detailed explanation for starting date, end date, number of shelters, and others.

Minor comment

Abstract section

: Detailed numbers of the DID analysis could be added in the Results section

: line 130 Figure 1. Current Figure 1 is hard to see and presents limited information. (Not sure why authors used different colored marks: Green Yellow, Red, Blue) I suggest authors to use map figure with additional information regarding earthquake epicenter.

: line 132 Authors mentioned similar characteristics, did not indicate what kinds of characteristics are similar. Is it population? Income? Number of hospitals? These factors should be similar across Group A and B, but they are not presented in the manuscript. Maybe supplementary table can be used here.

: Can’t see the Figure 2.

: Line 205-206 Not sure how the authors included gender and age information in the regression model instead of conducting stratification analysis. Please clarify (maybe using the equation?)

: Line 216-218 Other region-specific time-varying (e.g. monthly temperature for hospitalization analysis) variable can also include in the DID model. Not sure why authors did additional analyses using T-test.

: Figures 3 and 4 are not self-explanatory and hard to interpret. Please add figure titles to each figure. In addition, Figure 6b shows values below 0. Why is that?

7. PLOS authors have the option to publish the peer review history of their article (what does this mean? ). If published, this will include your full peer review and any attached files.

**Do you want your identity to be public for this peer review?** For information about this choice, including consent withdrawal, please see our Privacy Policy .

Reviewer #3: **Yes: ** Changwoo Han

---

## [Author Response · Author response to Decision Letter 2]

6 Feb 2025

Dear Editorial Office,

Dear Reviewer,

thank you for the opportunity to revise the manuscript. On behalf of all co-authors, I uploaded the response to reviewer's comments. For simplicity, I attached the point-by-point replies also here:

I agree with the previous reviewer 2’s comment, and changing the overall analyses using the DID framework seem reasonable. Major assumption of the DID analysis is parallel trend (as authors well indicated), and authors have evaluated it using statistical testing. However, basic evaluation of the assumption can be done with simple plotting of Group A and B (such as in Figure 4).

In the Figures 4a 4b 6b, we can clearly see the increase in hospitalization rate in Group A even before the earthquake event (2016 06, 07, 08, 09). How could authors explain these issues? Even a statistical testing for the common trend assumption shows marginally significant results, indicating that the potential difference of the trend before the earthquake event between Groups A and B. If authors shorten their analysis period for the common trend testing, I am bit worried that they might have significant p-values. I believe reasonable and clear explanations are needed for this issue before conducting DID analysis.

We thank the review for the valuable comment. The observed trend can be attributed primarily to the natural aging of the population in Group A, which led to a gradual increase in hospitalizations over time. This demographic shift, independent of the earthquake event, is a well-documented factor influencing hospitalization rates.

Additionally, regarding the peak observed in June 2016, we note that this period was characterized by an abnormal rise in temperatures due to a heat wave, which likely contributed to an increase in hospital admissions, particularly among the elderly population (https://www.ansa.it/english/news/general_news/2016/07/08/heatwave-in-italy-set-to-intensify_0a21cf60-582f-415a-8e7e-140b69448783.html). Heat waves are known to disproportionately affect vulnerable individuals, leading to higher rates of hospitalization for conditions such as dehydration, cardiovascular complications, and respiratory distress (e.g., Zhou M, Zhao G, Zeng Y, Zhu J, Cheng F, Liang W. Aging and cardiovascular disease: current status and challenges. Reviews in Cardiovascular Medicine. 2022 Apr 8;23(4):135).

We added a paragraph to provide explanations about this issue. Please, see lines 318-322.

More explanation for the biological plausibility of authors’ findings are needed. Current explanation in the introduction section focused mainly on the acute association (line 61-67, increase of stress right after the earthquake). Authors should explain why there are some lagged effects, and the effects of earthquake lasted for several months. Is it because of the long-lasting aftershocks? (BTW, I cannot see the figure 2 and hard for me to check the frequency of the aftershocks.)

We appreciate the reviewer’s insightful suggestion to further elaborate on the biological plausibility of the observed lagged effects of the earthquake on cardiovascular outcomes. As noted, our initial explanation focused primarily on the acute adrenergic response to the main tremor and immediate aftershocks. Nonetheless, it is known that earthquake-related stressors can exert prolonged effects on cardiovascular health.

A key factor contributing to the persistence of these effects is the prolonged seismic activity (up to January 2017), as the earthquake was followed by a series of aftershocks that continued for several months, as illustrated in Figure 2. This ongoing seismic activity likely sustained heightened stress levels in the affected population, leading to repeated activation of the sympathetic nervous system and prolonged exposure to elevated catecholamines. Such chronic stress responses have been associated with sustained increases in blood pressure, endothelial dysfunction, and an increased risk of cardiovascular events over time.

We clarified these aspects in the manuscript. Please, see lines 65-69.

To increase the possibilities of causal interpretation, I suggest authors to add some negative control analysis by using different intervention timing (some random point before the earthquake) or disease outcomes irrelevant to earthquake exposure.

Thank you for the valuable suggestion. Unfortunately, conducting such an analysis is not feasible due to data availability constraints, primarily related to privacy regulations that limit access to additional control data.

Moreover, it would be more appropriate to use years that are closer in time to the earthquake in order to minimize the influence of other confounding factors.

Notwithstanding, we believe that the robustness of our findings is supported by the consistency of the observed effect across multiple outcomes. Specifically, the impact of the earthquake on acute myocardial infarction is evident not only in hospitalization data but also in mortality records. Furthermore, our combined analysis of hospitalizations and deaths reinforces the plausibility of a true effect rather than a spurious association.

Precise referencing is needed. For example: line 59-60, 323-334, and many others.

Thank you for the precious suggestion. In response, we added the Wittstein et al. (2005) reference into the introduction to support the “fight or flight” adrenergic mechanism. Regarding the discussion section, the paragraph in question presents our own interpretation and primary implications of the study’s findings rather than a synthesis of previously published data; hence, we believe no additional reference is necessary there.

Current table 1 is not showing the results of the DID analysis. Please include tables showing the pre-earthquake rate, post-earthquake rate and DID estimates (main findings of this study). There should be tables corresponding to Figure 3-6 (please consider using the supplementary material).

We thank the reviewer for this suggestion. We have added to the Supplementary Material detailed tables showing the distribution of the pre‐ and post‐earthquake rates of mortality and hospitalizations.

We would like to clarify that Table 1 was not intended to display the results of the DID analysis; rather, it reports the overall rates for the two outcomes (hospitalization and death for AMI) as aggregated summary statistics, and it has been updated accordingly. In addition, because the overall AMI data (i.e., deaths + hospitalizations) were available on an annual basis only and comprised a limited number of observations, a DID approach was not feasible for these combined data. Instead, we employed an alternative analytic strategy using the chi‐square test to assess differences in these rates.

Please add more explanation for selecting different time frame for mortality and hospitalization. Authors presented it was based on the power calculation but not sure how this can be applied to the DID framework (Not familiar with DID studies using it). In addition, please change reference 15 16 to journal paper if it is possible. Line 121-124 needs more detailed explanation for starting date, end date, number of shelters, and others.

Thank you for raising these points. To clarify, the power calculation was performed using the relative risks observed in the pre‐ and post‐earthquake periods, and these calculations remain valid for our DID framework. In other words, the choice of different time frames for mortality and hospitalization outcomes was guided by these power considerations, ensuring that our analyses had sufficient statistical power to detect a meaningful effect in each case.

Regarding references number 15 and 16, these are primary institutional Italian sources that provided the necessary data for our expected values calculations. Unfortunately, journal papers reporting the same data are not available at this time.

Furthermore, we would like to clarify that the starting and ending dates of the study period were chosen to capture the window in which adverse post‐earthquake conditions—such as makeshift shelter accommodations (e.g., tents and cars) and extreme weather events like cold waves—were most likely to contribute to stress‐induced CCVD events. Although we agree that information on the number of shelters and similar specific parameters would further inform the context, such data were not available to us. In the manuscript, the examples provided (i.e., makeshift accommodations and cold waves) are intended to illustrate potential environmental stressors rather than to serve as quantitative inputs. We have now clarified in the revised text that while we cannot specify the exact number of shelters or related details due to data limitations, these examples underscore the plausibility of a prolonged period of adverse living conditions that could have increased the risk of CCVD events.

Detailed numbers of the DID analysis could be added in the Results section

Thank you for the valuable suggestion. We added the estimated values for the outcome whose difference in the pre- and post-period resulted to be statistically significant.

line 130 Figure 1. Current Figure 1 is hard to see and presents limited information. (Not sure why authors used different colored marks: Green Yellow, Red, Blue) I suggest authors to use map figure with additional information regarding earthquake epicenter.

Thank you for your precious suggestion. We revised Figure 1 by adding an earthquake epicenter. The different colored marks are needed to identify municipalities that belong to the several counties hit by the earthquake. Figure 1’s legend reports the list of municipalities for each marker.

line 132 Authors mentioned similar characteristics, did not indicate what kinds of characteristics are similar. Is it population? Income? Number of hospitals? These factors should be similar across Group A and B, but they are not presented in the manuscript. Maybe supplementary table can be used here.

The groups of municipalities A and B are similar in terms of demographic structure, socio-economic conditions and accessibility of care facilities. In the supplementary material, data and information on these aspects were added by conducting appropriate comparison tests

Can’t see the Figure 2.

Thank you for pointing out this issue. We replaced the older file and uploaded the final version of Figure 2.

Line 205-206 Not sure how the authors included gender and age information in the regression model instead of conducting stratification analysis. Please clarify (maybe using the equation?)

Actually, we considered age and gender as possible confounding factors. This aspect was specified in the methods section. Anyway, as suggested, we corrected the equation of the model including the two variables (age and gender). Please, see lines 205-206.

Line 216-218 Other region-specific time-varying (e.g. monthly temperature for hospitalization analysis) variable can also include in the DID model. Not sure why authors did additional analyses using T-test.

We thank the reviewer for the comment. In our study, we examined whether regional differences in temperature might affect the outcomes. However, our analysis of the temperature data revealed that there was no substantial variability in the monthly temperatures between areas of Group A and B. For this reason, temperature was not included as an additional time-varying covariate in the DID model. The t-test was specifically employed to test the difference between the mean temperatures in the two areas, and the results confirmed that there was no statistically significant difference.

Figures 3 and 4 are not self-explanatory and hard to interpret. Please add figure titles to each figure. In addition, Figure 6b shows values below 0. Why is that?

We thank the reviewer for these helpful suggestions. Regarding Figures 3 and 4, we note that their titles were included in the text; however, based on the reviewer’s comment, we have now revised the figures to include explicit titles and clear axis labels to enhance their self-explanatory nature. As for Figure 6b, the original model produced some values below 0. We have since revised the model by imposing a constraint that the rates cannot be less than 0. This update ensures that all output values are non-negative, consistent with the underlying epidemiological meaning. Please, see the revised Figures.

---

## [Editor Report · Decision Letter 2]

PONE-D-23-15690R2Mortality and admissions for cerebrovascular and cardiovascular diseases after the Accumoli-Amatrice 2016 earthquakePLOS ONE

Dear Dr. Nurchis,

Thank you for submitting your manuscript to PLOS ONE. After careful consideration, we feel that it has merit but does not fully meet PLOS ONE’s publication criteria as it currently stands. Therefore, we invite you to submit a revised version of the manuscript that addresses the points raised during the review process.

We apologize for the delay in the review, but it was really complex to identify the most suitable reviewers to verify the content of the document. Now, we kindly ask you to respond to the revisions you have made so that we can proceed quickly with the next steps.

We look forward to receiving your revised manuscript.

Kind regards,

Francesco Branda, Ph.D.

Academic Editor

PLOS ONE
---

## [Author Response · Author response to Decision Letter 3]

14 Apr 2025

I agree with the previous reviewer 2’s comment, and changing the overall analyses using the DID framework seem reasonable. Major assumption of the DID analysis is parallel trend (as authors well indicated), and authors have evaluated it using statistical testing. However, basic evaluation of the assumption can be done with simple plotting of Group A and B (such as in Figure 4).

In the Figures 4a 4b 6b, we can clearly see the increase in hospitalization rate in Group A even before the earthquake event (2016 06, 07, 08, 09). How could authors explain these issues? Even a statistical testing for the common trend assumption shows marginally significant results, indicating that the potential difference of the trend before the earthquake event between Groups A and B. If authors shorten their analysis period for the common trend testing, I am bit worried that they might have significant p-values. I believe reasonable and clear explanations are needed for this issue before conducting DID analysis.

We thank the review for the valuable comment. The observed trend can be attributed primarily to the natural aging of the population in Group A, which led to a gradual increase in hospitalizations over time. This demographic shift, independent of the earthquake event, is a well-documented factor influencing hospitalization rates.

Additionally, regarding the peak observed in June 2016, we note that this period was characterized by an abnormal rise in temperatures due to a heat wave, which likely contributed to an increase in hospital admissions, particularly among the elderly population (https://www.ansa.it/english/news/general_news/2016/07/08/heatwave-in-italy-set-to-intensify_0a21cf60-582f-415a-8e7e-140b69448783.html). Heat waves are known to disproportionately affect vulnerable individuals, leading to higher rates of hospitalization for conditions such as dehydration, cardiovascular complications, and respiratory distress (e.g., Zhou M, Zhao G, Zeng Y, Zhu J, Cheng F, Liang W. Aging and cardiovascular disease: current status and challenges. Reviews in Cardiovascular Medicine. 2022 Apr 8;23(4):135).

We added a paragraph to provide explanations about this issue. Please, see lines 318-322.

More explanation for the biological plausibility of authors’ findings are needed. Current explanation in the introduction section focused mainly on the acute association (line 61-67, increase of stress right after the earthquake). Authors should explain why there are some lagged effects, and the effects of earthquake lasted for several months. Is it because of the long-lasting aftershocks? (BTW, I cannot see the figure 2 and hard for me to check the frequency of the aftershocks.)

We appreciate the reviewer’s insightful suggestion to further elaborate on the biological plausibility of the observed lagged effects of the earthquake on cardiovascular outcomes. As noted, our initial explanation focused primarily on the acute adrenergic response to the main tremor and immediate aftershocks. Nonetheless, it is known that earthquake-related stressors can exert prolonged effects on cardiovascular health.

A key factor contributing to the persistence of these effects is the prolonged seismic activity (up to January 2017), as the earthquake was followed by a series of aftershocks that continued for several months, as illustrated in Figure 2. This ongoing seismic activity likely sustained heightened stress levels in the affected population, leading to repeated activation of the sympathetic nervous system and prolonged exposure to elevated catecholamines. Such chronic stress responses have been associated with sustained increases in blood pressure, endothelial dysfunction, and an increased risk of cardiovascular events over time.

We clarified these aspects in the manuscript. Please, see lines 65-69.

To increase the possibilities of causal interpretation, I suggest authors to add some negative control analysis by using different intervention timing (some random point before the earthquake) or disease outcomes irrelevant to earthquake exposure.

Thank you for the valuable suggestion. Unfortunately, conducting such an analysis is not feasible due to data availability constraints, primarily related to privacy regulations that limit access to additional control data.

Moreover, it would be more appropriate to use years that are closer in time to the earthquake in order to minimize the influence of other confounding factors.

Notwithstanding, we believe that the robustness of our findings is supported by the consistency of the observed effect across multiple outcomes. Specifically, the impact of the earthquake on acute myocardial infarction is evident not only in hospitalization data but also in mortality records. Furthermore, our combined analysis of hospitalizations and deaths reinforces the plausibility of a true effect rather than a spurious association.

Precise referencing is needed. For example: line 59-60, 323-334, and many others.

Thank you for the precious suggestion. In response, we added the Wittstein et al. (2005) reference into the introduction to support the “fight or flight” adrenergic mechanism. Regarding the discussion section, the paragraph in question presents our own interpretation and primary implications of the study’s findings rather than a synthesis of previously published data; hence, we believe no additional reference is necessary there.

Current table 1 is not showing the results of the DID analysis. Please include tables showing the pre-earthquake rate, post-earthquake rate and DID estimates (main findings of this study). There should be tables corresponding to Figure 3-6 (please consider using the supplementary material).

We thank the reviewer for this suggestion. We have added to the Supplementary Material detailed tables showing the distribution of the pre‐ and post‐earthquake rates of mortality and hospitalizations.

We would like to clarify that Table 1 was not intended to display the results of the DID analysis; rather, it reports the overall rates for the two outcomes (hospitalization and death for AMI) as aggregated summary statistics, and it has been updated accordingly. In addition, because the overall AMI data (i.e., deaths + hospitalizations) were available on an annual basis only and comprised a limited number of observations, a DID approach was not feasible for these combined data. Instead, we employed an alternative analytic strategy using the chi‐square test to assess differences in these rates.

Please add more explanation for selecting different time frame for mortality and hospitalization. Authors presented it was based on the power calculation but not sure how this can be applied to the DID framework (Not familiar with DID studies using it). In addition, please change reference 15 16 to journal paper if it is possible. Line 121-124 needs more detailed explanation for starting date, end date, number of shelters, and others.

Thank you for raising these points. To clarify, the power calculation was performed using the relative risks observed in the pre‐ and post‐earthquake periods, and these calculations remain valid for our DID framework. In other words, the choice of different time frames for mortality and hospitalization outcomes was guided by these power considerations, ensuring that our analyses had sufficient statistical power to detect a meaningful effect in each case.

Regarding references number 15 and 16, these are primary institutional Italian sources that provided the necessary data for our expected values calculations. Unfortunately, journal papers reporting the same data are not available at this time.

Furthermore, we would like to clarify that the starting and ending dates of the study period were chosen to capture the window in which adverse post‐earthquake conditions—such as makeshift shelter accommodations (e.g., tents and cars) and extreme weather events like cold waves—were most likely to contribute to stress‐induced CCVD events. Although we agree that information on the number of shelters and similar specific parameters would further inform the context, such data were not available to us. In the manuscript, the examples provided (i.e., makeshift accommodations and cold waves) are intended to illustrate potential environmental stressors rather than to serve as quantitative inputs. We have now clarified in the revised text that while we cannot specify the exact number of shelters or related details due to data limitations, these examples underscore the plausibility of a prolonged period of adverse living conditions that could have increased the risk of CCVD events.

Detailed numbers of the DID analysis could be added in the Results section

Thank you for the valuable suggestion. We added the estimated values for the outcome whose difference in the pre- and post-period resulted to be statistically significant.

line 130 Figure 1. Current Figure 1 is hard to see and presents limited information. (Not sure why authors used different colored marks: Green Yellow, Red, Blue) I suggest authors to use map figure with additional information regarding earthquake epicenter.

Thank you for your precious suggestion. We revised Figure 1 by adding an earthquake epicenter. The different colored marks are needed to identify municipalities that belong to the several counties hit by the earthquake. Figure 1’s legend reports the list of municipalities for each marker.

line 132 Authors mentioned similar characteristics, did not indicate what kinds of characteristics are similar. Is it population? Income? Number of hospitals? These factors should be similar across Group A and B, but they are not presented in the manuscript. Maybe supplementary table can be used here.

The groups of municipalities A and B are similar in terms of demographic structure, socio-economic conditions and accessibility of care facilities. In the supplementary material, data and information on these aspects were added by conducting appropriate comparison tests

Can’t see the Figure 2.

Thank you for pointing out this issue. We replaced the older file and uploaded the final version of Figure 2.

Line 205-206 Not sure how the authors included gender and age information in the regression model instead of conducting stratification analysis. Please clarify (maybe using the equation?)

Actually, we considered age and gender as possible confounding factors. This aspect was specified in the methods section. Anyway, as suggested, we corrected the equation of the model including the two variables (age and gender). Please, see lines 205-206.

Line 216-218 Other region-specific time-varying (e.g. monthly temperature for hospitalization analysis) variable can also include in the DID model. Not sure why authors did additional analyses using T-test.

We thank the reviewer for the comment. In our study, we examined whether regional differences in temperature might affect the outcomes. However, our analysis of the temperature data revealed that there was no substantial variability in the monthly temperatures between areas of Group A and B. For this reason, temperature was not included as an additional time-varying covariate in the DID model. The t-test was specifically employed to test the difference between the mean temperatures in the two areas, and the results confirmed that there was no statistically significant difference.

Figures 3 and 4 are not self-explanatory and hard to interpret. Please add figure titles to each figure. In addition, Figure 6b shows values below 0. Why is that?

We thank the reviewer for these helpful suggestions. Regarding Figures 3 and 4, we note that their titles were included in the text; however, based on the reviewer’s comment, we have now revised the figures to include explicit titles and clear axis labels to enhance their self-explanatory nature. As for Figure 6b, the original model produced some values below 0. We have since revised the model by imposing a constraint that the rates cannot be less than 0. This update ensures that all output values are non-negative, consistent with the underlying epidemiological meaning. Please, see the revised Figures.

---

## [Editor Report · Decision Letter 3]

PONE-D-23-15690R3Mortality and admissions for cerebrovascular and cardiovascular diseases after the Accumoli-Amatrice 2016 earthquakePLOS ONE

Dear Dr. Nurchis,

Thank you for submitting your manuscript to PLOS ONE. After careful consideration, we feel that it has merit but does not fully meet PLOS ONE’s publication criteria as it currently stands. Therefore, we invite you to submit a revised version of the manuscript that addresses the points raised during the review process.

The work, after the second revision, shows significant improvements in terms of scientific robustness and clarity, but still has some aspects to be refined for optimal publication. Strengths include better justification of the use of the Difference-in-Differences (DID) model, with tests for the assumption of parallel trends and more detailed explanations of the results; the addition of additional data that improve transparency; and effective clarifications on biological plausibility and pre-earthquake trends, supported by bibliographic references. The inclusion of age and gender in the model and the analysis of regional temperatures help reduce bias, while originality lies in filling a gap in the Italian literature with public health implications.Remaining critical points include limitations due to the use of aggregate data, which reduces granularity and increases the risk of ecological bias; the absence of a negative control analysis, which weakens causal inference despite the authors' justification; and the difference in pre-earthquake trends between groups, with marginal significance of the test for parallel trends (p=0.07 for AMI), which raises doubts about the validity of the DID.The lack of direct measurement of confounders such as socioeconomic status remains a limitation, as does the not always intuitive presentation of the figures and the absence of quantitative details on the selection criteria for control municipalities, which reduces reproducibility. The work compares with the international literature by being consistent, but less detailed in pathophysiological mechanisms and with less granular data than Japanese studies.In conclusion, while solid and deserving of publication, especially for the Italian context, we recommend adding an explicit statement of limitations, improving clarity of figures, and better discussion of pre-earthquake heterogeneity, possibly considering journals more specialized in disaster medicine or cardiology.

We look forward to receiving your revised manuscript.

Kind regards,

Emma Campbell, Ph.D

Staff Editor

PLOS One 

On behalf of 

<!--StartFragmentFrancesco Branda, Ph.D.

Academic Editor

PLOS One

---

## [Author Response · Author response to Decision Letter 4]

12 May 2025

The work, after the second revision, shows significant improvements in terms of scientific robustness and clarity, but still has some aspects to be refined for optimal publication. Strengths include better justification of the use of the Difference-in-Differences (DID) model, with tests for the assumption of parallel trends and more detailed explanations of the results; the addition of additional data that improve transparency; and effective clarifications on biological plausibility and pre-earthquake trends, supported by bibliographic references. The inclusion of age and gender in the model and the analysis of regional temperatures help reduce bias, while originality lies in filling a gap in the Italian literature with public health implications.

Thank you for your thoughtful and detailed assessment of our revised manuscript. We greatly appreciate your recognition of the improvements made regarding the scientific robustness and clarity of our work. We look forward to addressing the remaining aspects you have identified to further optimize our manuscript for publication.

Remaining critical points include limitations due to the use of aggregate data, which reduces granularity and increases the risk of ecological bias;

We thank the Editor for the valuable suggestion. The use of aggregated data was unavoidable because of restrictions due to privacy regulations concerning health data for academic researchers in our country. This limitation has been reported in the discussion and cannot be overcome. Please, see lines 410-419.

the absence of a negative control analysis, which weakens causal inference despite the authors' justification;

Thank you for the comment. This limitation has been added in the revised version. We also included data at country level showing that both mortality rates for CVD and hospitalization for AMI steadily decreased in the studied periods. Please, see lines 431-435.

and the difference in pre-earthquake trends between groups, with marginal significance of the test for parallel trends (p=0.07 for AMI), which raises doubts about the validity of the DID.

We thank the Editor for the precious comment. The test for parallel trends was largely not significant for CVD, both for mortality (p=0.15) and for hospitalization (p=0.75), while for AMI the p-values, even though not significant, are smaller: p=0.09 for mortality and p=0.07 for hospitalization. On the other hand, the graphical test did not indicate a clear parallel trend before the earthquake only for hospitalization rates for AMI, but the statistical tests supported the hypothesis that parallel trend assumption is not violated. As a result, we can be quite confident of the validity of the DID. For sake of clarity, we added a note on this aspect among the limits. Please, see lines 442-447

The lack of direct measurement of confounders such as socioeconomic status remains a limitation,

The quasi- experimental design adopted with the diff-in-diff analysis “in itself” guarantees the minimization of confounding . Notwithstanding, socioeconomic status could affect the outcome differently in the treatment and control groups. However, socioeconomic status was one of the criteria adopted to select the control group (see appendix, hence an indirect measurement of socioeconomic status was adopted. As suggested by the Editor, we included in the study’s limitations the failure to directly measure the potential confounding of socio-economic status. Please, see lines 406-409.

as does the not always intuitive presentation of the figures

Thank you for the comment. In the revised version we have tried to improve the presentation of the figures adding details and explanations. Please, see modified figures.

and the absence of quantitative details on the selection criteria for control municipalities, which reduces reproducibility.

Thank you for the observation. Quantitative details on the selection criteria were reported in the appendix and referred to in the text in the “study design” section. Briefly, the selection of control municipalities was conducted on the basis of demographic characteristics, socioeconomic level and access to hospital care services. Please, see the modified document.

The work compares with the international literature by being consistent, but less detailed in pathophysiological mechanisms and with less granular data than Japanese studies.

We thank the editor for this constructive feedback. We agree that our study, while consistent with international literature, provides less detailed pathophysiological mechanisms compared to some Japanese studies. We would like to note that the pathophysiological mechanisms were pointed out in the introduction section of our paper.

Japanese research on earthquake-related cardiovascular events has benefited from their extensive experience with seismic events and well-established disaster medicine infrastructure.

To address these limitations, we have expanded our discussion section to include more acknowledgment of the data granularity and pathophysiological mechanisms. We have added an explicit statement addressing these specific limitations in the manuscript, highlighting both the constraints of our data collection and the contextual differences between Italian and Japanese healthcare systems and disaster response infrastructure. Please, see lines 461-467.

In conclusion, while solid and deserving of publication, especially for the Italian context, we recommend adding an explicit statement of limitations, improving clarity of figures, and better discussion of pre-earthquake heterogeneity, possibly considering journals more specialized in disaster medicine or cardiology.

We appreciate the editor’s recognition of our work's value, particularly for the Italian context. We have addressed all three recommendations. First, we have expanded our limitations section to provide a more comprehensive and explicit statement of the study’s constraints. Second, we have improved the clarity of our figures by enhancing their resolution, standardizing the presentation format, and adding more descriptive captions. Third, we have added a discussion of pre-earthquake heterogeneity between the two groups, acknowledging potential confounding factors and how they might have influenced our results. Following your valuable suggestion, we have incorporated additional references from specialized journals in disaster medicine and cardiology to better contextualize our findings within these specific fields. While the pathophysiological mechanisms were already addressed in our introduction and a detailed exploration is beyond the scope of our study (which focuses on evaluating mortality and hospital admissions for CCVDs in the aftermath of the 2016 Accumoli-Amatrice earthquake), these references provide valuable comparative perspectives from international disaster responses, particularly enhancing the clinical relevance of our findings. Regarding pre-earthquake heterogeneity, we have addressed this important issue in our revised manuscript. Please, see lines 467-482.

---

## [Editor Report · Decision Letter 4]

Mortality and admissions for cerebrovascular and cardiovascular diseases after the Accumoli-Amatrice 2016 earthquake

PONE-D-23-15690R4

Dear Dr. Nurchis,

We’re pleased to inform you that your manuscript has been judged scientifically suitable for publication and will be formally accepted for publication once it meets all outstanding technical requirements.

Kind regards,

Francesco Branda, Ph.D.

Academic Editor

PLOS ONE
---

## [Editor Report · Acceptance letter]

PONE-D-23-15690R4

PLOS ONE

Dear Dr. Nurchis,

I'm pleased to inform you that your manuscript has been deemed suitable for publication in PLOS ONE. Congratulations! Your manuscript is now being handed over to our production team.

Kind regards,

on behalf of

Dr. Francesco Branda

Academic Editor

PLOS ONE